# Triboelectric Characterization of Colloidal TiO_2_ for Energy Harvesting Applications

**DOI:** 10.3390/nano10061181

**Published:** 2020-06-17

**Authors:** Erik Garofalo, Luca Cecchini, Matteo Bevione, Alessandro Chiolerio

**Affiliations:** 1Istituto Italiano di Tecnologia, Center for Sustainable Future Technologies, Via Livorno 60, 10144 Torino, Italy; erik.garofalo@iit.it (E.G.); matteo.bevione@iit.it (M.B.); 2Politecnico di Torino, Department of Electronics and Telecommunications, Corso Duca degli Abruzzi 24, 10129 Torino, Italy; lucacecchini26@gmail.com

**Keywords:** triboelectric effect, TENGs, energy harvesting, waste heat to power, colloid, liquid energy harvester

## Abstract

Nowadays, energy-related issues are of paramount importance. Every energy transformation process results in the production of waste heat that can be harvested and reused, representing an ecological and economic opportunity. Waste heat to power (WHP) is the process of converting the waste heat into electricity. A novel approach is proposed based on the employment of liquid nano colloids. In this work, the triboelectric characterization of TiO_2_ nanoparticles dispersed in pure water and flowing in a fluorinated ethylene propylene (FEP) pipe was conducted. The idea is to exploit the waste heat to generate the motion of colloidal TiO_2_ through a FEP pipe. By placing an Al ring electrode in contact with the pipe, it was possible to accumulate electrostatic charges due to the triboelectric effect between the fluid and the inner pipe walls. A peristaltic pump was used to drive and control the flow in order to evaluate the performances in a broad fluid dynamic spectrum. The system generated as output relatively high voltages and low currents, resulting in extracted power ranging between 0.4 and 0.6 nW. By comparing the power of pressure loss due to friction with the extracted power, the electro-kinetic efficiency was estimated to be 20%.

## 1. Introduction

According to a recent study [1], the world energy consumption has followed an increasing trend since the first industrial era. In 2040, world energy consumption is expected to exceed 200,000 TWh/year, and the industry sector (expected to grow by an 18% from 2015 to 2040) will be responsible for more than 50% [2]. For this reason, energy-related issues including the exploitation of resources, the cost transformation as well as awareness about utilization, are of paramount importance. Furthermore, energy transformation must deal with non-ideal thermodynamic process, resulting in a distributed production of waste heat that can be harvested and reused, providing an attractive opportunity for an emission-free and cost-effective resource [3]. Energy harvesting can be defined as the process wherein the sources such as mechanical load, vibrations, temperature gradients, heat, light, salinity gradients, wind, etc., are scavenged and converted to obtain relatively small levels of power in the nW-mW range [4]. More in detail, waste heat to power (WHP) is the process of converting the heat discarded by an existing process in electricity. The main source of waste heat is the industrial sector, but also in domestic, automotive, and aerospace fields, energy harvesting is an important milestone of the environmentally friendly industries and processes of the future since, currently, low enthalpy thermal loads with a temperature below 250 °C are discarded [5]. Heat rejected into the environment is one of the largest sources of clean, fuel-free, and inexpensive energy available. Many WHP methods have been already addressed, based mainly on thermodynamic cycles and solid-state devices, such as thermoelectric generators (TEGs). Usually, thermal waste is managed installing large cogeneration plants, involving moving parts and having thermodynamic limits. On the other end, thermoelectric devices partially solve those problems but suffer the drawbacks of employing expensive and eventually toxic materials, reaching optimal performances in often narrow temperature ranges. Mixed effects enhanced at the nanoscale are, nowadays, less employed but very promising, even in other research fields where the employment of multifunctional nanomaterials has highlighted their remarkable properties [6]. Magnetocaloric and thermomagnetic machines but also pyroelectric [7,8] and triboelectric devices, are very appealing, especially when featuring simple, scalable and emission-free design. 

A novel WHP approach is based on the employment of liquid nano colloids. Colloids are complex condensed matter systems lying at the boundary between completely homogeneous systems such as solutions and completely heterogeneous systems such as suspensions [9]. Usually they are defined as stable suspensions of nanoparticles (NPs) in a carrier fluid, in a liquid or gaseous form. They are classified according to the aggregation state of their components, the dispersant (carrier phase), and the dispersoid (suspended phase). These fluids are characterized by a suspended phase with the ability to confer specific physical properties according to the nature of the NPs (magnetic, thermal, optical, electronic, etc.,), without interfering with the liquid nature of the solution. The other advantage is that within the limit of stability, the global system can be considered as single-phase, since the mutual interaction among the particles and the molecules of the carrier liquid are balanced by the Brownian motion (induced by the thermal effects) and by the physicochemical properties of a dispersant. However, colloids feature non-trivial collective properties and can undergo phase transitions, for instance, flocculation, occurring when the dispersoids coalesce; gelation, occurring when the dispersant changes viscosity; and Coulomb explosion [9], occurring when electrostatic repulsion produces a sudden pressure increase. 

In this work, the triboelectric characterization of TiO_2_ nanoparticles dispersed in pure water and flowing in a fluorinated ethylene propylene (FEP) pipe has been conducted. The idea is to exploit the waste heat to generate the motion of colloidal TiO_2_ through a FEP pipe, governed by natural or Rayleigh-Bénard convection processes [10]. Placing an Al ring electrode in contact with the pipe, it is possible to accumulate electrostatic charges due to the triboelectric effect between the fluid and the inner pipe walls. As typical for triboelectric phenomena, the accumulated charges generate as output relatively high voltages and low currents, respectively in the range of 300–600 mV and 400–900 pA. The triboelectric effect is defined as a contact-induced electrification in which a material becomes electrically charged after it is contacted with a different material through friction [11]. It is possible to experience this effect in everyday life, since it is the cause of electrostatic phenomena. However, the mechanism behind triboelectrification is still under debate between a solid-state approach and a more holistic one based on an engineering model. When two different materials come into contact, a chemical bonding is established between the surfaces of the materials (adhesion) and if the process is slow enough, owing to electrons, holes, or ion/molecules a flow of charges occurs in order to reach the thermal equilibrium, leading the electrochemical potential of the two materials to equalize. The sign of the charges carried by a material depends on its relative polarity in comparison to the material with which it is in contact [11]. 

The triboelectric nanogenerator (TENG) is the first device exploiting contact electrification to efficiently convert mechanical energy into electricity [12] and it has been systematically studied to instantaneously drive hundreds of light-emitting diodes (LEDs) [13], and charge a lithium-ion battery for powering a wireless sensor and a commercial cell phone [14]. Recently, new devices have been developed in order to collect energy from other environmental sources, such as wind [15], human motion [16], ocean waves, and tides [17]. Until nowadays, TENGs have been designed employing solid materials whose optimal performances are obtained under dry conditions [18]. However, triboelectricity has also been demonstrated when liquids flow through pipes or polymeric films made of dielectric materials, from which a new application of TENG can derive [18]. As examples, voltages up to 300 mV have been observed when deionized water (DIW) flows through a 1-m long rubber tube [19], and surface charge densities of 4.5 C/cm^2^ have been measured on water droplets injected by a polytetrafluoroethylene (PTFE) tip [20]. In fact, water plays an important role in the electrification of materials. A particular kind of contact electrification (flow electrification) has been reported in different studies [21,22], where insulating liquids cause electrification of electronic components or determine shock problems in petroleum pipeline hazards. The key point in order to understand how it is possible to exchange charges in a water solution is to determine what is the position of water in the so called “triboelectric series”, which sort the materials by the relative polarity. An interesting analysis provided by Burgo et al. [23] gives a complete outlook on the electrification processes in pure water, stating that ion-partitioning near solid-liquid interfaces (hydroxyl adsorption of water-hydrophobic surfaces) is fundamental in material electrification, while an example of water-based TENGs using a Ti-mesh, operating in single electrode mode (SEM) is provided in the work of Park et al. [24]. The SEM configuration is that of a parallel plate capacitor where only one electrode is wired, the other being left floating. The idea of employing titanium oxide powder derives from the work of Lin et al. [25], in which a rationally designed solid-state TENG has been developed by using the contact electrification between a PTFE thin film and a layer of TiO_2_ nanomaterial (nanowire and nanosheet) array. The as-developed TENG has been systematically studied and demonstrated as a self-powered nanosensor toward catechin detection, with output voltage and current density increased by a factor of 5.0 and 2.9 respectively, demonstrating the good triboelectric match between PTFE and TiO_2_ [25]. The concept has been strengthened from the review of Matsusaka et al. [26], where the triboelectric properties of powders in general are highlighted. Recently, with the advent of low power electronics (recovery systems, environmental monitoring, Internet of Things, etc.,), triboelectric devices have found to be used as a renewable energy source, to harvest energy from several types of mechanical energy, including vibrational, rotational, wind, and tidal wave energy. The important features of these devices consist in their ease of fabrication, stability, and high-energy conversion efficiency [24]. So, the right functionalization of TENGs is important in order to build a marketable device for recovery applications, even from unexplored energy sources such as waste heat. The present work is included in a broader research project related to CAS (colloidal autonomous system) concept [27], in which colloidal devices, properly protected from the external environment by a deformable skin, are used in order to realize a revolutionary system for the purpose of liquid-robotics. With respect to the well-established heavy, steel-based robotics, this kind of technology can be used in harsh environment applications, as space exploration, or the deepest zones of remote lakes and oceans, or post-disaster rescue searches, or even in surgery operations. From an energetic point of view, this autonomous system needs to be coupled with an energy harvesting/storage module in an efficient way. A multiphase colloidal system using different recovery mechanisms, such as the triboelectric effect, is here tested to achieve this aim [7].

## 2. Materials and Methods

Inspired by the work of Ravelo et al. [19], a simple experimental setup was designed, in order to characterize the triboelectric effect of a water-based colloidal solution. The experimental setup consists of a cylindrical FEP pipe in which the motion of a colloidal solution of TiO_2_ nanoparticles (Evonik Degussa, Essen, Germany) and DIW is forced by a peristaltic pump (Cole-Parmer GmbH, Wertheim, Germany); the region addressed to the extraction of the electrostatic charge was placed half-way of the pipe in order to reduce the fluid dynamics disturbance associated to the progressive reduction of the section of the pipe, starting from the pump up to the extraction region. Finally, an aluminum ring in SEM configuration was installed externally to the pipe for accumulating the charges developed by triboelectric effect and being able to perform the capacitive characterization of the system. A picture of the setup is shown in Figure 1.

Four different suspensions were prepared with a volume concentration of 0.5, 1, 2, and 4% of 30 nm titanium dioxide NPs. The TiO_2_ powder whose purity is extremely high (TiO_2_ content > 99.5%) with a composition of 40% rutile and 60% anatase, was dispersed in DIW (purchased from Carlo Erba, Milan, Italy, featuring an electrical resistivity of 12.8 MΩ/cm), by means of a 30 min ultrasonic bath (Bransonic^®^ M, Branson Ultrasonics, Danbury, CT, USA, operating at 40 kHz). In Figure 2, two samples of the suspensions are shown, which feature an excellent stability toward sedimentation avoiding electrostatic aggregation of the NPs, even without the need of adding surfactants (that would have hindered electronic transfer phenomena) or of balancing pH.

The colloidal suspension flows through the FEP pipe with a 10 mm inner diameter (ID) and 12 mm outer diameter (OD). The choice of the materials stems out from their triboelectric properties [11] and for its chemical compatibility with water-based suspensions. The charging process that occurs in polymers is not well understood as in metal but there are several consistent charging patterns observed. In particular, insulators and organic polymers can be arranged in the triboelectric series from those that charge most positive, like nylon, to those that charge most negative, like the halogenated polymers [28]. FEP is defined as a negative element in the triboelectric series, also due to its fluorinated groups in the polymeric structure [29,30]. Since water tends to charge positively in presence of electronegative elements, FEP is the perfect candidate for the study of triboelectric effects. The sizing of the pipe, instead, was driven by fluid dynamics and economic considerations, taking into account only cylindrical pipes because of their well-established and simple physics. In fact, to directly associate electrostatic phenomena to the fluid dynamics behavior of the colloid within the pipe, a Poiseulle flow is preferred, in order to place the capacitive electrode in a region featuring a fully developed velocity profile.

In the suspension of titania in DIW with a volume concentration of 1%, the density *ρ*, and dynamic viscosity μ, are respectively 1.03 × 10^3^ kg/m^3^ and 9.1 × 10^−4^ kg/m s, the latter calculated using the Einstein linear relation [31]. Considering the associated Reynolds number:(1)$$Re=\frac{\rho \;d\;v}{\mu }$$
where *d* (m) is the internal diameter and *v* (m/s) the mean fluid velocity, it was possible to design an internal diameter of 10 mm, given a maximum mean velocity inside the pipe of 4.25 cm/s, which is a reasonable value for the velocity of the colloid in a real device, supposing a motion triggered and governed by natural or Rayleigh-Bénard convection processes. 

The peristaltic pump employed is the Ismatec MCP equipped with a rotor system which limits the maximum internal diameter of the silicone pipe (suitable for the peristaltic motion) to 3.2 mm. Operating at the maximum rotational speed, the pump can generate a maximum flow rate of 100 mL/min for each channel, of which only two are needed to comply with the fluid dynamic requirements, following the formula: (2)$$Q=\langle v\rangle \;D^{2}\frac{\pi }{4}$$
where ⟨v⟩ (m/s) is the mean velocity of the fluid parallel to the unit vector normal to the surface *A* (m^2^), which is defined as *D^2^ π/*4 in the case of cylindrical pipe. Using a set of hydraulic adapters, it was possible to move the colloidal solution from a beaker placed close to the pump through the FEP pipe with the minimum fluid dynamic disturbance. 

The extraction system consists of an aluminum electrode anchored externally to the FEP pipe, installed for the single electrode mode characterization. The choice of aluminum was conducted considering the triboelectric coupling with FEP, since Al tends to accumulate positive charges on its surface. The reference electrode is chosen as the same ground of the measurement instruments, in order to guarantee the repeatability of the measurements. The Al ring is a screw clamp, having an ID of 12 mm an OD of 28 mm and a length of 9 mm. 

The electrical behavior of the device was observed setting the peristaltic pump at five different fixed velocities, in the range between 0.8 and 4.3 cm/s, and analyzing the corresponding values of voltage potentials and currents generated at the Al electrode by the colloid motion. In fact, the capacitive characterization of the triboelectric effect consists in the evaluation of the charge accumulation on the aluminum ring electrode, by means of open-circuit voltage and short-circuit current measurements, V_OC_ and I_SC_, performed respectively by the Keithley 2635A—System SourceMeter^®^ and the Keithley 4200-SCS Semiconductor Characterization System coupled with a Keithley 4225-RPM Remote Amplifier (Tektronix, Inc., Beaverton, OR, USA). The choice of the instruments was driven by their accuracy in the expected measurements range. All the instrumentation is connected to a PC with LabVIEW 2019 software installed (National Instruments, Austin, TX, USA), to control the data acquisition system. In order to verify the effective action of the triboelectric effect, a preliminary analysis on the time evolution of voltage and current was performed, varying the operational condition of the pump. Furthermore, in order to reduce the vibrational noise due to the peristalsis motion of pump, a fast Fourier transform (FFT) analysis was conducted with the purpose of observing the existence of vibrational modes besides the fundamental one, due to direct current source. A similar analysis was not performed for the voltage measurements since the sensitivity of the instrument is not sufficient to capture the phenomenon. To better understand the direct link between the peristalsis motion and the frequency modes generation, a Gaussian fit was conducted on the FFT analysis of the current measurements. Then, after this preliminary analysis, the Savitzky-Golay fitting was applied to reduce the vibrational noise given by the pulsed motion of the fluid. Finally, to evaluate the voltage and current generated by the triboelectric effect a subset of data was selected, neglecting the initial transient and focusing on the steady state periods, and then filtering these data using a moving average filter, with respect to their relative offset values (when the pump is off). The modeling of the system considers a RC equivalent circuit, where the resistive component is associated to the conductivity of the solution, the capacitive component to the geometrical properties of the electrodes and to the insulating properties of the polymer interface and, finally, the current flowing in the electrodes proportional to the velocity in the pipe.

## 3. Results

### 3.1. Feasibility Analysis

In order to verify the effective action of the triboelectric effect when the fluid is moving through the pipe, a preliminary analysis of the time evolution of voltage and current measurements was conducted, switching on and off the peristaltic pump using DIW as working fluid. The results of this analysis are reported in Figure 3.

From Figure 3, a direct relation between the variation of the flow velocity into the pipe and the voltage generation on the ring electrode, with respect to the ground state, is observed. In particular, when the pump is switched on (off), a steep increase (decrease) of the output voltage can be noticed, as typical for a RC circuit, where the resistive component is associated to the conductivity of the solution, the capacitive component to the geometrical properties of the electrodes and to the insulating properties of the polymer interface and finally, the current flowing in the electrodes proportional to the velocity in the pipe [21]. A complete quantitative analysis is presented in Section 3.4.

### 3.2. Data Filtering

In order to measure V_OC_ and I_SC_ at equilibrium, a data subset, associated to the final part of the measurement, after the characteristic transient time, was used (raw data of both open circuit voltages and short circuit currents are available in the Appendix A file). Then, a moving average filter (first-order polynomial Savitzky-Golay filter) was applied, with a fixed filtering window of 51 samples. The choice of this filtering window is related to the minimization of the error band. It is important to note that three samples per second are acquired during the voltage measurement while, for the current, the sampling rate is five times larger because of the specifics of the instrumentation. An example of data subset for V and I is shown in Figure 4, with the relative polynomial filter applied.

### 3.3. Frequency Analysis

Considering the high noise level of current measurements, with respect to constitutive conditions of these experiments, a fast Fourier transform analysis was performed on the experimental values of the current, in order to detect vibrational modes (besides the fundamental one related to the triboelectric source) and reduce the vibrational noise due to peristalsis. An example is given in Figure 5a, where the frequency of DIW flowing at 2.6 cm/s was analyzed and filtered. To better understand the direct relation between the peristaltic motion and the frequency mode generation, a Gaussian fit was applied, as shown in Figure 5b. 

This analysis highlighted important information about the frequency shift, power and full width at half maximum (FWHM) of the vibrational modes when a change of the flow velocity is introduced, with respect to the fundamental vibrational mode. The complete frequency analysis in the cases of DIW and 1% TiO_2_-water mixture are reported in Table 1 and Table 2.

By looking at Table 1 and Table 2 a decrease of the first mode center frequency is evident, occurring when the mean flow velocity increases. The explanation could be related to the mechanical frequency response of the pump-pipe-bench system, considering that the pipe is anchored to the latter. In the case of titanium dioxide solution, Table 2, the data show that the variation of mechanical properties of the fluid (in particular density and viscosity) gives rise to a reduction of the noise power.

### 3.4. Capacitive Characterization

Open-circuit voltage and short-circuit current measurements were performed setting five different flow velocities inside the pipe: 0.85, 1.7, 2.6, 3.4, and 4.3 cm/s. In the cases of DIW, 0.5% and 1% of TiO_2_ in water, results are provided in Figure 6.

As it can be seen from Figure 6a, the potential of Al electrode shows an increasing trend with respect to the mean flow velocity, reaching a plateau between v = 2.6 cm/s and up to v = 4.3 cm/s, corresponding approximately to V_OC_ = 450 mV. Considering the current values, a fluctuation is evident with a current peak at v = 3.4 cm/s corresponding to a short circuit current I_SC_ = 920 pA. The maximum ideal output power is obtained simply multiplying V_OC_ by I_SC_. The maximum power is generated at 3.4 cm/s, with P_max_ = 402 pW obtained with a reference surface of S = 4.14 × 10^−4^ m^2^, given by the geometrical properties of the aluminum ring electrode. Therefore DIW-based setup is able to output a power per unit area of P_A_ = 971 nW/m^2^. Taking into consideration Figure 6b, the triboelectric effect of TiO_2_-water suspension is not strong as it might be expected. The extracted potential is negative up to v = 4 cm/s, the current extremely noisy and faintly negative, reaching at maximum -70 pA. The most relevant value is observed at v = 4.25 cm/s with a V_OC_ = 298 mV, I_SC_ = 72 pA, P_max_ = 21 pW, and P_A_ = 52 nW/m^2^. Considering Figure 6c, an increase of magnitude of both voltage and current, starting from lower velocities (and up to v = 2.6 cm/s), is found. Nevertheless, further increasing the mean flow velocity, an inversion of this trend occurs in voltage measurements. This phenomenon can be related to a reduction of the friction between the solution and the pipe wall, consequent to a reduction in the flow velocity in correspondence of the walls. In fact, neutral tracers added to the colloid have shown a steep decrease in the peripheral flow velocity when exceeding 2.6 cm/s. The peak in the power generated is found at v = 2.6 cm/s, where V_OC_ = 575 mV and the associated I_SC_ = 950 pA, corresponding to P_max_ = 546 pW and P_A_ = 1.319 μW/m^2^, with a three-fold increase (an improvement of 287%) with respect to the case of DIW at the same velocity (P_max_ = 190 pW). 

## 4. Discussion

In order to complete the analysis of this triboelectric energy harvester it is interesting to understand the amount of energy due to the friction between the colloidal solution and the inner pipe walls converted into electrical energy. Theoretically the triboelectric phenomenon is described as phonon induced [32]. In fact, the viscous friction forces generate in part vibrations in the polymeric structure of the pipe (phonon generation), in part electron-transfer (triboelectric effect) and a remaining part of dissipation in the form of heat. In Figure 7a schematic representation of the macroscopic phenomenon is shown. Furthermore, other approaches [11] define the number of charge carriers extracted by triboelectric effect as a function of energetic compatibility (affinity, Fermi level, etc.,) which influences directly the triboelectric series. This means that a system constituted of same colloid flowing at the same velocity would produce electrical power with a different efficiency according to the different material employed for the pipe (FEP, PTFE, PVC, and so on). From these considerations and adopting a macroscopic approach the authors found it appropriate to estimate the electro-kinetic efficiency comparing the electrical power extracted by the harvester and the power loss due to the friction forces at the fluid-wall interface. First, considering only the portion of the pipe corresponding to the length of the Al electrode L (m) (where the charge accumulation is possible), following the Darcy-Weisbach equation [33], the pressure drop ∆p (Pa) due to viscous effects was calculated:(3)$$∆p=\rho f\frac{L}{D}\frac{\langle v\rangle {\;}^{2}}{2}$$
where ⟨v⟩ is the mean velocity (m/s) and *f* the Darcy friction factor expressed by the Colebrook-White equation [34], for a conduit flowing completely full of fluid as:(4)$$\frac{1}{\sqrt{f}}=-2\log\left(\frac{\varepsilon }{3.7D}+\frac{2.51}{Re\sqrt{f}}\;\right)$$
where ε is the pipe roughness (m) equal to 0.1 μm and *D* (m) is the pipe inner diameter and Re the associated Reynolds number.

To evaluate *P_friction_*, the power due to the pressure loss *∆p* was multiplied by the volumetric flow rate Q (m^3^/s). Finally, in the situation with the highest electrical power extracted (0.546 pW), the electro-kinetic conversion efficiency η was estimated to be 20%, comparing *P_friction_* with *P_electrical_* even if it could be further increased matching the direction of dipoles formed [35]:(5)$$\eta =\frac{P_{electrical}}{P_{friction}\;}=\frac{V_{oc}\;I_{sc}}{Q\;∆p}$$

Although it is out of the scope of this paper, it would be interesting to analyze how the harvesting system affects the fluid motion and consequently the system performances. In particular, how the accumulation of electrostatic charges in correspondence of the electrode, representing a region of electrostatic attraction/repulsion, influences the pressure loss. If this attraction/repulsion were modeled as an increase/decrease of roughness, a variation of 1 µm in the roughness generates a variation of 0.1% in the system efficiency. Another option would be to investigate the direct effect on the flow of the distribution of electrostatic charges in the electrode region [36]. 

## 5. Conclusions

In conclusion, a device for electro-kinetic energy harvesting, based on a liquid state nano colloid (titania dispersed in DIW) and its triboelectric interaction with the pipe walls, made of fluorinated ethylene propylene (FEP) were described. Although the work is based on the use of FEP, the same approach could be extended to several other polymeric materials having a broader employment in general piping systems such as polyvinyl chloride (PVC), since they are defined as negative element in the triboelectric series [11,28]. Electrostatic charges accumulated were collected using an Al electrode, while motion speed was controlled using a peristaltic pump. The simple motion of the colloid generated as output open circuit voltages in the range of 0.6 V and closed-circuit currents in the range of 1 nA. The physical phenomena taking place in the device and the proposed expression for the conversion efficiency, estimated to be 20%, were discussed. In this work the ratio between the FEP pipe roughness (about 100 nm) and the TiO_2_ nanoparticles diameter (30 nm) was ≈ 3; future works could investigate how this ratio affects the system performances, since it could be a driving parameter in designing colloidal triboelectric energy harvester. Furthermore, controlling the ferroelectric domains of the pipe wall could optimize and amplify the electrostatic charge distribution at the level of the extraction electrodes. Considering the originality of this work, it was difficult to compare and evaluate the goodness of the obtained results but the idea is to further investigate this triboelectric system with the employment of different nano materials. The authors found the work of Pjesky et al. (2009) [37] interesting, where nano particles of TiO_2_ and MgO were used to charge a PTFE tribocharger by means of a powder disperser showing that titania gained a higher net charge.

Finally, a novel approach for energy harvesting was presented, with several opportunities and fields of applications such as, as mentioned, in CAS (colloidal autonomous system) for liquid-robotics but also in automotive applications where, as an example, the car cooling systems could be coupled with a triboelectric energy harvesting systems. 

## Figures and Tables

**Figure 1 nanomaterials-10-01181-f001:**
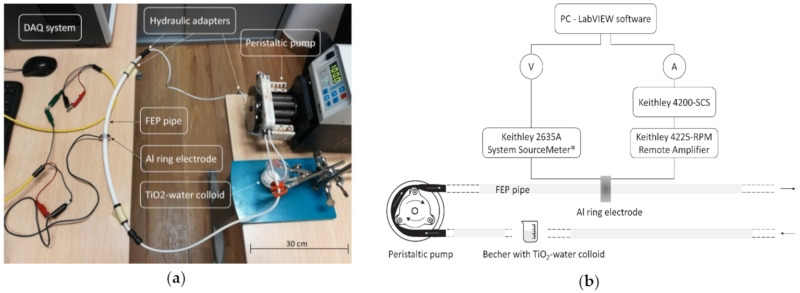
(**a**) Picture of the experimental setup with a focus on the capacitive electrode; (**b**) schematic of the experimental setup where all the components are highlighted.

**Figure 2 nanomaterials-10-01181-f002:**
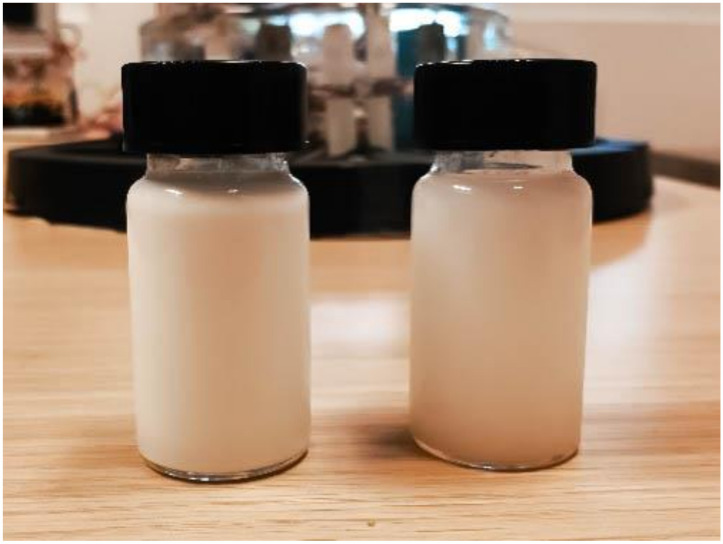
Suspensions of TiO_2_ in DIW with volume concentration of 4% (left) and 2% (right).

**Figure 3 nanomaterials-10-01181-f003:**
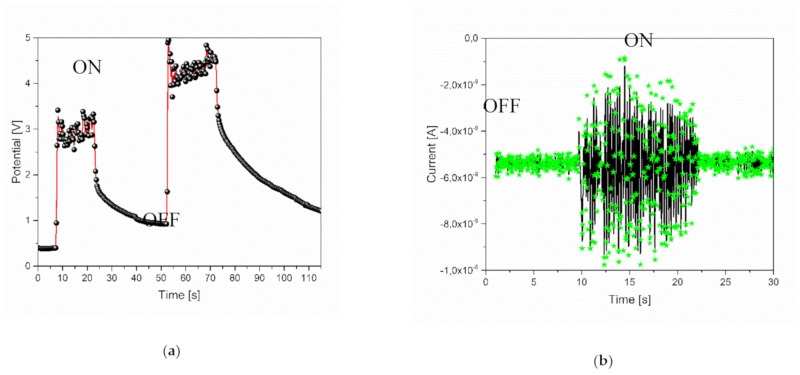
Evidence of triboelectric effect from: (**a**) voltage-time and (**b**) current-time analysis of 1% TiO_2_ solution.

**Figure 4 nanomaterials-10-01181-f004:**
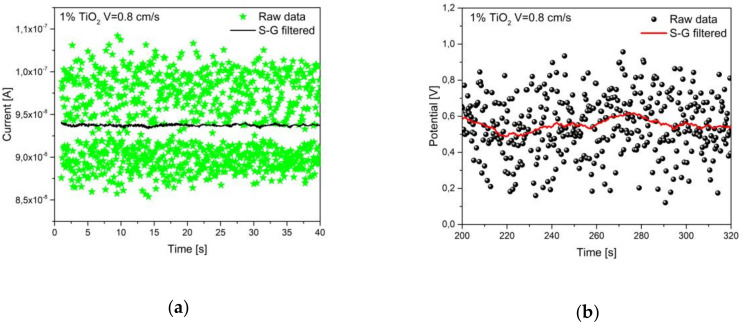
I (**a**) and V (**b**) measurements in time (1% titania suspension, 0.8 cm/s speed). Comparison between raw and filtered data.

**Figure 5 nanomaterials-10-01181-f005:**
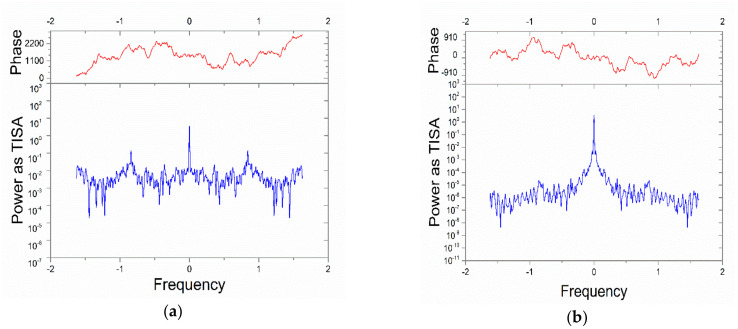
Frequency analysis on the current measurements before (**a**) and after filtering (**b**) associated to the current response of DIW at v = 2.6 cm/s.

**Figure 6 nanomaterials-10-01181-f006:**
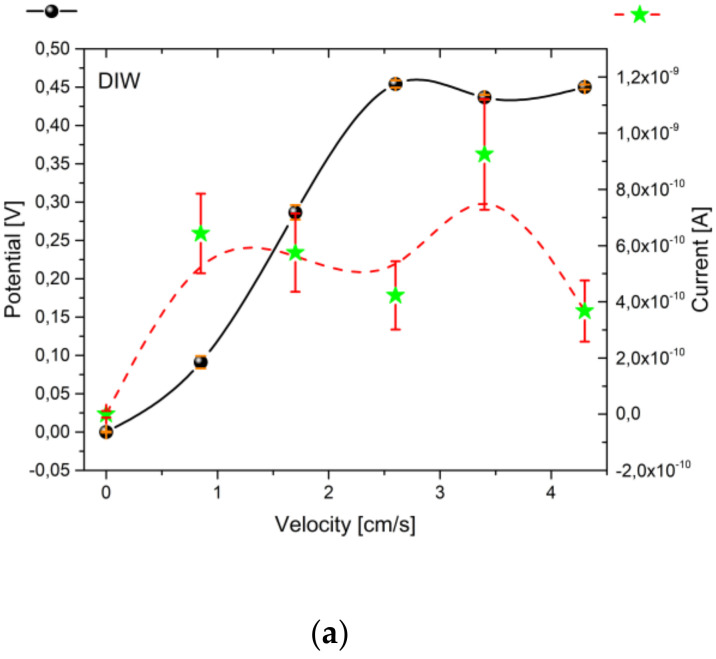
Open-circuit voltage and short-circuit current characterization in function of the mean velocity of DIW (**a**), 0.5% (**b**) and 1% (**c**) TiO_2_-water suspension.

**Figure 7 nanomaterials-10-01181-f007:**
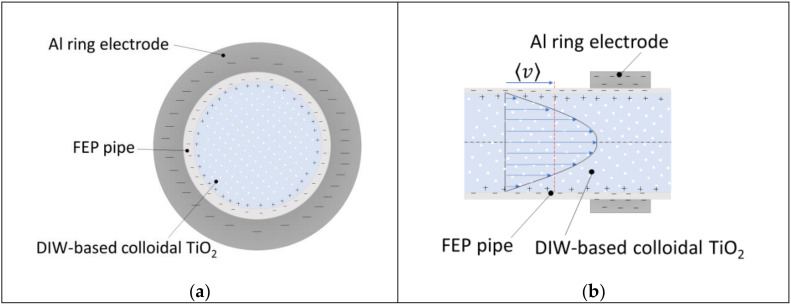
Representation of triboelectric phenomenon in cross section (**a**) and lateral section (**b**) view.

**Table 1 nanomaterials-10-01181-t001:** Frequency analysis of the first mode associated to the current measurement at different mean velocities in the case of DIW.

Fluid Velocity (cm/s)	Relative Amplitude (%)	Mode Frequency Peak (Hz)	FWHM (Hz)
0.8	25	5.8	0.23
1.7	11	4.2	0.93
2.6	15	1.5	0.61
3.4	9	1.4	0.21
4.3	6	1.9	1.22

**Table 2 nanomaterials-10-01181-t002:** Frequency analysis of the first mode associated to the current measurement at different mean velocities in the case of 1% TiO_2_-water mixture.

Fluid Velocity (cm/s)	Relative Amplitude (%)	Mode Frequency Peak (Hz)	FWHM (Hz)
0.8	10	3.1	0.33
1.7	8	2.2	0.26
2.6	6	2.1	0.74
3.4	4	2.1	0.56

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
