# Peer review of "Triboelectric Characterization of Colloidal TiO2 for Energy Harvesting Applications"

_nanomaterials, 2020, doi:10.3390/nano10061181_

Round 1

Reviewer 1 Report

Manuscript by Garofalo reported the harvesting of power from waste heat using TiO2 nano colloids  in a Fluorinated Ethylene Propylene (FEP) pipe with 20 % of  electro-kinetic efficiency. The results have potential interest. Hence manuscript can be accepted for publication after minor revision.

  1. Characterization of TiO2 should be included for proper understanding of synthesized material.  
  2. A comparative table should be included showing the performance of TiOwith other nanomaterials.

Author Response

The authors wish to thank by hearth the two reviewers for their positive comments, and constructive criticisms. We hope that they will find reasonable responses in the following letter, and a properly revised manuscript (all changes highlighted in yellow).

Reviewer#1

Manuscript by Garofalo reported the harvesting of power from waste heat using TiO2 nano colloids in a Fluorinated Ethylene Propylene (FEP) pipe with 20 % of electro-kinetic efficiency. The results have potential interest. Hence manuscript can be accepted for publication after minor revision.

  1. Characterization of TiO2 should be included for proper understanding of synthesized material.

For running the experiments whose results are shown in the paper, a TiO2 powder was purchased from Evonik Degussa and not synthetized in lab. The powder purity is high, with TiO2 content > 99.5% (based on ignited material) and a composition of 40% rutile and 60% anatase, having an average particle size (evaluated by using the SEM) of 30 nm. Some of the missing details have been added to section 2.

  1. A comparative table should be included showing the performance of TiOwith other nanomaterials.

We appreciate the interests of the reviewer for this comparative analysis, but unfortunately, giving the originality of this work a very few papers have been found in literature. Pjesky et al. (2009) [37] employed TiO2 and MgO nano powders to charge a PTFE tribocharger by means of a powder disperser showing that titania gained a higher net charge. For these reasons, the authors aim to further investigate this novel approach employing different nano materials. In section 5 a few sentences have been added: “Considering the originality of this work, it was difficult to compare and evaluate the goodness of the obtained results but the idea is to further investigate this triboelectric system with the employment of different nano materials. The authors found interesting the work of Pjesky et al. (2009) [37] where nano particles of TiO2 and MgO were used to charge a PTFE tribocharger by means of a powder disperser showing that titania gained a higher net charge.”

Reviewer 2 Report

Waste Heat to Power is the process of converting the waste heat in electricity. In this study, authors have introduced a novel approach based on the employment of liquid nano colloids. More specifically, they have experimented with TiO2 nanoparticles dispersed in pure water, and flowing in a Fluorinated Ethylene Propylene (FEP) pipe. The paper is well written and the results are significant. The novelty of the study is there. I have minor comments that need to be addressed before publication

1) The paper does not discuss the applicability of this technique to general piping systems. Here in where I live, the PVC pipe is the name of the game. Can this method be applied? The bigger picture is the broader impact. Can the authors add a few sentences on broader impact of their work? I have a feeling that they are not thinking as broad as they should

2) This can not be a big deal to some reviewers, but it is too much. It is distracting to have font changes in the middle of the section. Please be consistent from the professionalism end of the spectrum.

3) The important figure of the paper has error bars. I applaud the authors in including the error bars. It may be my oversight but I didn't come across statistical analysis though. It would be good to include the statistically significant or not when comparing two sets of experimental data points.

4) The conclusion section seems recycled from the rest of the paper. Please take a closer look at opportunities to rewrite some of the sentences. 

Author Response

Reviewer#2

Waste Heat to Power is the process of converting the waste heat in electricity. In this study, authors have introduced a novel approach based on the employment of liquid nano colloids. More specifically, they have experimented with TiO2 nanoparticles dispersed in pure water, and flowing in a Fluorinated Ethylene Propylene (FEP) pipe. The paper is well written and the results are significant. The novelty of the study is there. I have minor comments that need to be addressed before publication

1) The paper does not discuss the applicability of this technique to general piping systems. Here in where I live, the PVC pipe is the name of the game. Can this method be applied? The bigger picture is the broader impact. Can the authors add a few sentences on broader impact of their work? I have a feeling that they are not thinking as broad as they should

We thank the reviewer for the remarkable suggestion to extend the application analysis in view of commercial exploitation. Starting from this suggestion, we broadened our understanding on the application of this technique to other polymeric materials, especially PVC, and found positive signs. The conclusion section (5) has been improved with a few sentences: “Although the work is based on the use of FEP, the same approach could be extended to several other polymeric materials having a broader employment, also in general piping systems, such as polyvinyl chloride (PVC), since they are defined as negative element in the triboelectric series [11, 28].”

2) This can not be a big deal to some reviewers, but it is too much. It is distracting to have font changes in the middle of the section. Please be consistent from the professionalism end of the spectrum.

The font has been checked and uniformed throughout the entire work.

3) The important figure of the paper has error bars. I applaud the authors in including the error bars. It may be my oversight but I didn't come across statistical analysis though. It would be good to include the statistically significant or not when comparing two sets of experimental data points.

We have included a new file, the Supporting Information, where raw data are shown, for the three liquids (DIW and the two TiO2 suspensions) and for one velocity (3.4 cm/s). Raw data consist of two curves for each material (the open circuit voltage and closed circuit current), from each of the curve one single point with its associated standard deviation was extracted, and included in the plot of Figure 6.

4) The conclusion section seems recycled from the rest of the paper. Please take a closer look at opportunities to rewrite some of the sentences. 

We found very useful this particular suggestion. We added some sentences in the conclusion section (5) to better explore and suggest the opportunity of this novel energy harvesting system: “In this work a novel and original approach for energy harvesting has been presented, with several opportunities and fields of applications such as, as mentioned, in CAS (Colloidal Autonomous System) for liquid-robotics but also in automotive where, as an example, the car cooling systems could be coupled with triboelectric energy harvesting systems.
